# ARE/Nrf2 Transcription System Involved in Carotenoid, Polyphenol, and Estradiol Protection from Rotenone-Induced Mitochondrial Oxidative Stress in Dermal Fibroblasts

**DOI:** 10.3390/antiox13081019

**Published:** 2024-08-21

**Authors:** Aya Darawsha, Aviram Trachtenberg, Yoav Sharoni

**Affiliations:** Department of Clinical Biochemistry and Pharmacology, Faculty of Health Sciences, Ben-Gurion University of the Negev, Beer Sheva 8410500, Israel; ayadar@post.bgu.ac.il (A.D.); aviramtr@post.bgu.ac.il (A.T.)

**Keywords:** mitochondrial dysfunction, dermal fibroblasts, rotenone, reactive oxygen species (ROS), matrix metalloproteinase (MMP), collagen, antioxidant response element/Nrf2 (ARE/Nrf2), tomato extract, rosemary extract, estradiol

## Abstract

Skin aging is associated with the increased production of mitochondrial reactive oxygen species (mtROS) due to mitochondrial dysfunction, and various phytonutrients and estrogens have been shown to improve skin health. Thus, the aim of the current study was to examine damage to dermal fibroblasts by chemically induced mitochondrial dysfunction and to study the mechanism of the protective effects of carotenoids, polyphenols, and estradiol. Rotenone, a Complex I inhibitor, caused mitochondrial dysfunction in human dermal fibroblasts, substantially reducing respiration and ATP levels, followed by increased mitochondrial and cytosolic ROS, which resulted in apoptotic cell death, an increased number of senescent cells, increased matrix metalloproteinase-1 (MMP1) secretion, and decreased collagen secretion. Pre-treatment with carotenoid-rich tomato extracts, rosemary extract, and estradiol reversed these effects. These protective effects can be partially explained by a cooperative activation of antioxidant response element (ARE/Nrf2) transcriptional activity by the protective compounds and rotenone, which led to the upregulation of antioxidant proteins such as NQO1. To determine if ARE/Nrf2 activity is crucial for cell protection, we inhibited it using the Nrf2 inhibitors ML385 and ochratoxin A. This inhibition markedly reduced the protective effects of the test compounds by diminishing their effect to reduce cytosolic ROS. Our study results indicate that phytonutrients and estradiol protect skin cells from damage caused by mtROS, and thus may delay skin cell senescence and improve skin health.

## 1. Introduction

Reactive oxygen species (ROS) play an essential role in regulating various cell functions and biochemical processes. Increased levels of ROS can be generated from environmental sources such as exposure to UV radiation and air pollutants [1], as well as from endogenous sources such as mitochondrial respiration and NADPH oxidases [2]. Oxidative stress, defined as an imbalance between ROS production and antioxidant activity [3], induces damage to nucleic acids, proteins, and lipids and the modulation of signaling pathways [4], activating inflammatory processes that may lead to harmful processes such as neurodegeneration [5] and carcinogenesis [6], and to accelerated aging [7,8].

Mitochondrial respiration and oxidative phosphorylation, which is the main source of ATP in aerobic cells, are a major source of ROS [9]. During energy production through the electron transport chain, some electrons “leak” to oxygen [10], forming superoxide radicals (O_2_^−.^) primarily from Complexes I and III [10]. Superoxide radicals are converted to hydrogen peroxide (H_2_O_2_) by superoxide dismutases in and out of the mitochondria, and both superoxide and hydrogen peroxide can exit the mitochondria and increase cytosolic ROS [11]. Mitochondrial dysfunction and reduced electron transport activity lead to decreased ATP production and an increased production of superoxide radicals [12], which are involved in the pathophysiology of diseases including Parkinson’s [13], as well as in the aging process [12], as suggested by Harman’s free radical theory of aging [14].

Mitochondrial dysfunction is involved in both chronological and photo-aging of the skin [15]. Aged skin is characterized by mutations of mitochondrial DNA (mtDNA), damaged mitochondria, and oxidative stress in both the dermal and epidermal layers [16]. Mitochondrial dysfunction that was induced in IMR-90 human fibroblasts by the depletion of mtDNA, or by using inhibitors of mitochondrial respiration, rotenone, or antimycin A, was shown to induce a senescent phenotype, as determined by several markers, including the increased activity of senescence-associated beta-galactosidase (SA-β-gal) [17]. The senescent cell phenotype includes the inhibition of proliferation, morphological changes, and the senescence-associated secretory phenotype. Oxidative stress causes skin damage due to the stimulation of inflammatory processes [18], the induction of cellular apoptosis [19], the increased expression of metalloproteinases (MMPs) which degrade extracellular matrix proteins [20,21], and reduced collagen synthesis [21], resulting in decreased skin elasticity and wrinkles. By increasing cellular antioxidant activities, oxidative stress should be lowered, thus reducing skin damage and skin aging. A recent study has shown in dermal fibroblasts that rotenone, a Complex I inhibitor, increased oxidative markers, induced apoptosis, increased MMP-1 expression, and decreased collagen 1a1 expression [22]. This study suggests that rotenone can be used as a model for skin cell aging as it increased the levels of SA-β-gal, a marker for senescent and aging cells.

Epidemiological and clinical studies have shown that a diet rich in fruits and vegetables promotes skin health [23]. The phytonutrients of carotenoids and polyphenols were found to reduce skin damage [24,25] and to protect dermal fibroblasts by eliminating oxidative stress [26]. It has been suggested that this reduction in ROS is due to the activation of the antioxidant defense mechanism [27]. Previous studies demonstrated that carotenoids [28,29] and polyphenols [30,31] activate the antioxidant response element/Nrf2 (ARE/Nrf2) transcription system in several cell types. Additional mechanisms for the protective effects of phytonutrients involve the modulation of signaling pathways which lead to skin damage [32]. Several studies suggest that phytonutrients modulate these signaling pathways, resulting in reduced skin cell death [33] and inflammation [30], and in changes in MMPs and collagen levels [26,32].

In menopause, when estradiol levels decrease, skin health is impaired [34]. This impairment can be partially overcome by estrogens, which are known to increase collagen and elastin levels and reduce MMPs that lead to enhanced skin elasticity [34,35]. Estrogens also increase skin cell viability and proliferation, leading to enhanced skin thickness [35]. These estrogen effects may be related to a reduction in ROS [34], which can result from the estrogen-induced expression of antioxidant enzymes [36] or by the indirect activation of the ARE/Nrf2 transcription system [37].

The aim of the current study was to investigate the effect of endogenous oxidative stress in primary human dermal fibroblasts that results from rotenone-induced mitochondrial dysfunction, and to examine the protective role of carotenoids, polyphenols, and estradiol. Rotenone disrupts the electron transport chain and reduces mitochondrial respiration, thereby increasing mitochondrial oxidative stress [38]. As rotenone was shown to also increase SA-β-gal activity [22], the exposure of dermal fibroblasts to rotenone can be used as a model for mitochondria-generated ROS (mtROS)-induced skin cell aging and can allow for testing strategies to slow down this process and to identify mechanisms that are involved in its acceleration and attenuation. The effect of rotenone and the treatment compounds was examined on apoptotic cell death, MMP-1 and pro-collagen 1a1 secretions, mitochondrial and cytosolic ROS levels, mitochondrial function, and SA-β-gal activity as a marker of cellular senescence. The activation of the ARE/Nrf2 transcription system by rotenone and by dietary compounds and estradiol, and its inhibition by specific inhibitors were examined to suggest the role this system has in protecting dermal fibroblasts from the damage caused by mtROS.

## 2. Materials and Methods

### 2.1. Materials

Tomato extracts (Lycomato^TM^) and rosemary extract were a gift of Lycored Ltd., (Beer Sheva, Israel). The red tomato extract, prepared through the ethyl acetate extraction of tomato pulp, contained 6% lycopene, other tomato carotenoids (phytoene and phytofluene above 1%, beta-carotene above 0.2%), and additional fat-soluble tomato components such as natural tocopherols (above 1.5%) and phytosterols (1.1–2.5%). The remainder were triacylglycerols (70–72%), monoacylglycerols (8–9%), and phospholipids (7–8%). The golden tomato extract contained phytoene 6.7%, phytofluene 1.9%, zeta-carotene 2.0%, β-carotene 0.14%, lycopene 0.1%, tocopherols 3.1%, and phytosterols 1.0%. The remainder were triacylglycerols (71%), monoacylglycerols (8%), and phospholipids (6%). Carotenoids were dissolved in tetrahydrofuran (THF, containing 0.025% butylated hydroxytoluene as an antioxidant; Sigma-Aldrich (Rehovot, Israel) and solubilized in cell culture medium, and their final concentration was measured as described previously [29,39,40]. The rosemary extract was prepared through extraction with 80% ethanol. Its composition was only partially determined to contain carnosic acid (20.2%) and carnosol (2.5%) as the main polyphenols. When presenting results with these extracts, the concentrations are given as the concentration of the main active component in each extract—lycopene for the red tomato extract, phytoene for the golden tomato extract, and carnosic acid for rosemary extract. 17β-estradiol and rotenone were purchased from Sigma-Aldrich (Rehovot, Israel). The rosemary extract and estradiol were dissolved in ethanol, and rotenone was dissolved in DMSO. Hanks’ solution and 1M HEPES buffer were purchased from Biological Industries (Beth Haemek, Israel). Dulbecco’s modified Eagle’s medium (DMEM) and dextran-coated charcoal-treated fetal bovine serum (DCC-FBS) were acquired from Capricorn Scientific (Ebsdorfergrund, Germany). Rotenone was purchased from Sigma-Aldrich (Rehovot, Israel). Ochratoxin A (OTA) was purchased from Fermentek (Jerusalem, Israel). ML385 was purchased from Cayman Chemical (Ann Arbor, MI, USA).

### 2.2. Cell Culture

Normal human dermal fibroblasts (NHDFs) were purchased from PromoCell GmbH (Heidelberg, Germany). The cells were grown in PromoCell fibroblast growth medium 2, according to the manufacturer’s instructions, in a humidified atmosphere at 37 °C in 5% CO_2_. Before each experiment, the cells were depleted of steroid hormones by maintaining them for 3–5 days in phenol red-free DMEM, supplemented with 10% DCC-FBS. This medium was used throughout all of the experiments because it does not contain steroid hormones or any other compound with estrogenic activity, such as phenol red.

### 2.3. Determination of Cell Number, and Secretion of MMP-1 and Procollagen 1a1

NHDF cells were seeded in 96-well plates at a density of 5 × 10^3^ cells/well in DMEM-DCC-FBS medium, and 24 h later, the cells were pre-incubated with phytonutrients or estradiol for another 24 h. Vehicle-treated control cells were incubated with the relevant amounts of solvents used in a particular experiment, which had no effect on the measured parameters. The medium was then replaced with one containing the treatment compounds, with or without 1 µM rotenone, and incubated for an additional 48 h. This concentration was determined after titrating the effect of rotenone on cell number in the range of 10^−8^ M to 10^−5^ M (Appendix A). There were no significant differences in the reduction in cell number by these concentrations, but the reproducibility of the results was better at 1 µM. Thereafter, the medium was removed and frozen for the analysis of secreted proteins, and the cell number was determined using a crystal violet assay (Sigma-Aldrich, Rehovot, Israel), according to the manufacturer’s instructions. MMP-1 and pro-collagen 1a1 protein levels in cell culture supernatants were quantified through ELISA using Human Total MMP-1 DuoSet and Human Pro-Collagen 1a1 DuoSet ELISA kits (R&D Systems, Inc., Minneapolis, MN, USA), according to the manufacturer’s instructions. Optical density was measured using a VERSAmax tunable microplate reader (Molecular Devices, Menlo Park, CA, USA). The results of the cell number, MMP-1, and pro-collagen measurements were calculated as a percentage of the values obtained in control cells, which were treated with a vehicle without rotenone.

### 2.4. Assessment of Apoptosis

Cells were seeded in 6-well plates at 3 × 10^5^ cells/well, and 24 h later, they were pre-incubated with the phytonutrients or estradiol for 24 h. The medium was then replaced with one containing the treatment compounds, with or without rotenone, and incubated for an additional 48 h. The cells were washed in PBS and stained with annexin V using an Annexin V-FITC/7-AAD Apoptosis Detection Kit (Biogems, Cat# 62700-50, Chai Wan, Hong Kong), according to the manufacturer’s protocol. The percentages of apoptotic cells were determined using flow cytometry (Beckman Coulter Inc., Brea, CA, USA). For each analysis, 10,000 events were recorded, and the data were processed using Kaluza software, version 2.1 (Beckman Coulter). As a positive control, cells were incubated with 1.25 µM staurosporin for 24 h. Annexin V-positive cells were considered to be in the apoptotic phase.

### 2.5. ApoLive-Glo Multiplex Assay for Cell Viability and Apoptosis

Because rotenone is known to affect mitochondrial respiration, we confirmed the results from a crystal violet assay (Sigma-Aldrich) using a method not based on mitochondrial activity. To this end, we used an ApoLive-Glo Multiplex Assay Kit (Promega, Madison, WI, USA) according to the manufacturer’s instructions. The level of the viable cells was measured using a fluorogenic substrate of the ApoLive-Glo Multiplex Kit. Furthermore, the apoptosis results obtained by annexin V staining were confirmed by measuring caspase-3/7 activation, a key biomarker of apoptosis, using the luminogenic caspase-3/7 substrate of the ApoLive-Glo Multiplex Kit. Both cell viability and caspase-3/7 activity were detected using a SpectraMax Paradigm plate reader (Molecular Devices Co., Sunnyvale, CA, USA).

### 2.6. Determination of Cytosolic Levels of Reactive Oxygen Species

Cytosolic ROS levels were determined through 2’,7’-dichlorofluorescin diacetate staining (DCFH-DA, Sigma-Aldrich). Intracellular ROS oxidizes this probe to a highly fluorescent compound, DCF. Cells were seeded in 96-well plates at 10^4^ cells/well, and 24 h later, they were pre-incubated with the phytonutrients or estradiol for another 24 h. The medium was then replaced with one containing the treatment compounds, with or without rotenone, and incubated for an additional 4 h. The cells were washed with Hanks’ solution containing 10 mM HEPES buffer, pH = 7.4, and, subsequently, stained with 40 μM DCFH-DA for 30 min at 37 °C in the dark. Fluorescence was detected using a SpectraMax Paradigm plate reader (Molecular Devices Co.). As a positive control, DCFH-DA-loaded cells were treated with 0.5 mM H_2_O_2_ for 10 min. Untreated and unstained cells served as a negative control.

### 2.7. Determination of Mitochondrial Levels of Reactive Oxygen Species

Mitochondrial ROS levels were stained with MitoSOX Red superoxide indicators. The oxidation of the MitoSOX reagent by mitochondrial superoxide produces red fluorescence. Cells were seeded in 96-well plates at 10^4^ cells/well, and 24 h later, they were pre-incubated with the phytonutrients or estradiol for 24 h. The medium was then replaced with one containing the treatment compounds, with or without rotenone, and incubated for an additional 90 min. The cells were washed with Hanks’ solution containing 10 mM HEPES buffer, pH = 7.4, and, subsequently, stained with 5 μM MitoSOX for 30 min at 37 °C in the dark. The cells were analyzed in a SpectraMax Paradigm plate reader (Molecular Devices Co.). As a positive control, MitoSOX-loaded cells were treated with 5 µM TBHP for 10 min. Untreated and unstained cells served as a negative control.

### 2.8. RNA Extraction, cDNA Synthesis, and Gene Expression Analysis (RT-qPCR)

Cells were seeded at 10^5^ cells/well, and 24 h later, they were pre-incubated with test agents for 24 h. The medium was then replaced with one containing the treatment compounds, with or without rotenone, and incubated for an additional 48 h. An RNA extraction kit (GENEzol^TM^ TriRNA Pure Kit+DNASE I; Geneaid, New Taipei City, Taiwan) was used to extract and purify the total RNA from cell cultures. The RNA was quantified using a micro-volume spectrophotometer (NanoDrop; Wilmington, DE, USA). mRNA was then reverse-transcribed to cDNA using a reverse transcriptase kit (qScript cDNA synthesis kit; QUANTA Biosciences; Gaithersburg, MD, USA), and quantitative cDNA amplification was performed through real-time PCR (StepOne Real-Time PCR System; ThermoFisher Scientific; Wilmington, DE, USA) using an advanced master mix (ThermoFisher). A TaqMan assay was used for the gene expression analysis of collagen type 1 (COL1A1) (Hs00164004_m1), with actin (Hs99999903_m1) as the housekeeping gene (ThermoFisher). Each treatment was studied with three biological replicates.

### 2.9. Mitochondrial-Respiration Parameters

Seahorse XF-96 Extracellular Flux (Agilent Technologies, Inc., Santa Clara, CA, USA) was used to measure key parameters of the mitochondrial function by measuring the oxygen consumption rate (OCR) and glycolysis by measuring the extracellular acidification rate (ECAR), according to the manufacturer’s instructions. The modulators included in this assay were oligomycin, carbonyl cyanide-4 (trifluoromethoxy), phenylhydrazone (FCCP), rotenone, and antimycin. Cells were seeded in 96-well XF plates at 10^4^ cells/well, and 24 h later, they were pre-incubated with the phytonutrients or estradiol for 24 h. The medium was then replaced with one containing the treatment compounds with rotenone and incubated for an additional 24 h. Each sample was assayed in triplicate. OCR parameters were calculated using Wave software version 2.6 and an Agilent Seahorse XFe Analyzer (version 2.6.3.5). The results were normalized to the living cell number in each well using DAPI staining.

### 2.10. Cellular ATP Levels

Cells were seeded in 96-well plates at 10^4^ cells/well, and 24 h later, they were pre-incubated with test agents for 24 h. The medium was then replaced with one containing the treatment compounds, with or without rotenone, and incubated for an additional 48 h. ATP cellular levels were determined using a CellTiter-Glo Luminescent Cell Viability Assay Kit (Promega, Madison, WI, USA) according to the manufacturer’s instructions. The cells were incubated with ATP assay buffer at room temperature for 10 min. Luminescence was then measured in quadruplicate using a SpectraMax Paradigm plate reader (Molecular Devices Co., Sunnyvale, CA, USA).

### 2.11. Preparation of Whole Cell Lysates and Western Blotting

Cells were seeded in 6-well plates at 10^5^ cells/well, and 24 h later, they were pre-incubated with test agents for 24 h. The medium was then replaced with one containing the treatment compounds, with or without rotenone, and incubated for an additional 48 h. The cells were lysed in a buffer containing 1% (*v*/*v*) Triton X-100 at 4 °C. 20 μg of each sample were separated using SDS-PAGE, and electroblotted onto nitrocellulose membranes. The membranes were blocked with 5% nonfat milk in Tris-buffered saline containing 0.5% Tween 20 (TBST) for 2 h, and incubated with primary antibodies overnight at 4 °C, followed by 1 h incubation with the secondary antibody, Anti-Mouse IgG (H+L) (cat# 5450-0011, Sera Care, Milford, MA, USA). The following primary antibodies were used: NQO1 (Cat# SC-271116, 1:500), with β-actin as the internal loading control (Cat# SC-47778, 1:500), both purchased from Santa Cruz Biotechnology Inc. (Santa Cruz, CA, USA). Protein bands were visualized using Western Lightning^TM^ Chemiluminescence Reagent Plus (PerkinElmer Life Sciences, Inc., Boston, MA, USA). The absorbance of each band was determined using an Image Quant LAS 4000 system (GE Healthcare Life Science, Little Chalfont, UK), and the optical density of each band was determined using ImageQuant^TM^ TL software v 8.0 (GE Healthcare Life Science). The optical density values for each band were normalized to the β-actin.

### 2.12. Transient Transfection and ARE/Nrf2 Reporter Gene Assay

NHDF cells were seeded in 24-well plates at 10^5^ cells/well, and 24 h later, they were transfected using jetPEI reagent (Polyplus Transfection, Illkrich, France). The cells were then rinsed once with serum-free medium, followed by the addition of 0.45 mL of medium and 50 µL of a mixture containing DNA and jetPEI reagent at a charge ratio of 1:5. The total amount of DNA was 0.25 µg, containing 0.2 µg 4xARE reporter construct that was kindly provided by Dr. M. Hannink (University of Missouri-Columbia, Columbia, MO, USA) and 0.05 µg Renilla luciferase (P-RL-null) expression vectors, which served as an internal transfection standard, and was purchased from Promega (Madison, WI, USA). These conditions were found to be optimal for the dermal fibroblasts. The cells were then incubated for 6 h at 37 °C. Next, the medium was replaced with DMEM-DCC-FBS plus the test compounds, and the cells were incubated for an additional 16 h. Cell extracts were prepared for the luciferase reporter assay (Dual Luciferase Reporter Assay System, Promega), according to the manufacturer’s instructions, and luminescence was determined using a Turner Biosystems luminometer (Sunnyvale, CA, USA).

### 2.13. Determination of SA-β-Galactosidase Activity

Senescent cells were detected by their SA-β-gal activity using a qualitative, semi-quantitative staining method and a fluorescence quantitative method. For the staining method, cells were seeded in 6-well plates at 2 × 10^5^ cells/well, and 24 h later, the cells were pre-incubated with test agents for 24 h. The medium was then replaced with one containing the treatment compounds, with or without rotenone, and incubated for an additional 48 h. Then, the cells were washed with PBS and stained with a solution containing the X-gal substrate, according to the manufacturer’s instructions, using a Senescence Cells Histochemical Staining Kit (Sigma-Aldrich) for 24 h. Thereafter, the wells were imaged using a PrimoVert microscope. Blue-stained, senescent cells were counted using ImageJ software version 1.54 g within a 2.2 mm^2^ area. Etoposide (25 µM), known to induce cell senescence, was used as a positive control. For the fluorescence method, cells were seeded in 24-well plates at 2 × 10^5^ cells/well and treated as above. Then, the cells were incubated for about an hour with the substrate, which turns fluorescent by SA-β-gal activity, using a Senescence Assay Kit (Beta Galactosidase, Fluorescence; Abcam, Cambridge, UK). Subsequently, the level of positive cells was determined through flow cytometry (Beckman Coulter Inc.). Approximately 10,000 cells were sampled from each test, and the analysis was performed using Kaluza software, version 2.1 (Beckman Coulter).

### 2.14. Statistical Analysis

All experiments were conducted three to seven times in triplicate. The number of independent experiments is indicated in the legends to figures. Statistically significant differences between two experimental groups were determined using a one-way ANOVA with Dunnett’s multiple comparison post hoc analysis. A *p*-value of less than 0.05 was considered statistically significant. The statistical analyses were performed using GraphPad Prism 9.0 software (Graph-Pad Software, San Diego, CA, USA).

## 3. Results

### 3.1. Mitochondrial and Cytoplasmic ROS Levels in Dermal Fibroblasts Are Increased by Rotenone Treatment and Reduced by Tomato and Rosemary Extracts, and Estradiol

The inhibition of mitochondrial respiration by rotenone should result in oxidative stress. Thus, to confirm that rotenone increases mitochondrial and cytoplasmic oxidative stress, ROS levels were detected using Mitosox and DCFH fluorescent probes, respectively. The average geometric means of fluorescence intensities (MFI) were measured 2 h after adding rotenone for Mitosox and 4 h for DCFH. Rotenone treatment increased ROS levels by six-fold in the mitochondria (Figure 1a) and four-fold in the cytosol (Figure 1b). The increase in mitochondrial ROS levels was reduced by about 70% when the cells were pre-incubated with tomato and rosemary extracts and estradiol (Figure 1a), and the cytosolic ROS level was reduced by these compounds almost to the basal level (Figure 1b). It was then important to assess if these changes in ROS levels were associated with parallel changes in cell function.

### 3.2. Rotenone Triggers Apoptotic Cell Death of Dermal Fibroblasts, Which Can Be Reduced by Tomato and Rosemary Extracts and Estradiol

To investigate whether the rotenone-induced rise in ROS level results in cell death, and whether tomato and rosemary extracts and estradiol can protect the cells, the number of dermal fibroblasts was measured using crystal violet staining after treatment with 1 µM rotenone for 48 h, with or without pre-treatment for 24 h with the protective compounds. Rotenone reduced the cell number to 40–50% of the control number. Pre-incubation with tomato or rosemary extracts or estradiol before exposure to rotenone in the presence of the treatment compounds resulted in a significant and dose-dependent increase in cell number (Figure 2a–d). The concentration of the treatment compounds chosen for further experiments were those giving the maximal effects for rosemary extracts (10 μM carnosic acid), red tomato extract (10 μM lycopene), and golden tomato extract (40 μM phytoene). The concentration chosen for estradiol (10 nM) was the lowest one showing a significant effect. Although the effect with 100 nM of estradiol was somewhat higher than at 10 nM, this concentration was avoided as it was shown to cause additional, non-receptor-dependent effects [41,42].

Cell number, measured through crystal violet staining, was compared to cell viability that was detected using an ApoLive-Glo Multiplex Viability Assay. Rotenone treatment for 48 h decreased both cell number and cell viability by 50–60% (Figure 2e,f, respectively). To reveal if the reduction in cell number and viability was associated with apoptotic cell death, apoptosis was analyzed in cells treated as above, using two methods: annexin V fluorescence and caspase 3 activity with an ApoLive-Glo^TM^ Multiplex Caspase-Glo^®^ Assay. Typical flow cytometric data of one experiment are shown in Figure 2g. About 10% of the cells were apoptotic in the control cells, whereas the percentage of annexin V-positive cells increased remarkably upon exposure to rotenone, with the average increase being about 6-fold (Figure 2h, gray bar). A similar increase of about 5-fold was found by measuring caspase 3 activity—a key biomarker of apoptosis (Figure 2i, gray bar). Pre-incubation for 24 h with tomato or rosemary extracts or estradiol, at the concentrations determined in Figure 2a–d, before exposure to rotenone in the presence of the treatment compounds, resulted in a significant reduction in apoptosis (Figure 2h,i) and an increase in cell number and viability (Figure 2e,f).

### 3.3. Cellular Senescence Is Increased following Exposure to Rotenone and Reduced by Dietary Compounds and Estradiol

SA-β-gal is one of the most widely used biomarkers for senescent cells. The effects of rotenone and the treatment compounds on the number of cells containing the enzyme which became stained blue were investigated. Etoposide, a known cellular senescence inducer, was used as a positive control and resulted in about a 6-fold increase in the number of stained cells (Figure 3a,b). In cells treated with rotenone, a higher increase of approximately 11-fold was observed (Figure 3a,b). Pre-treatment with the tomato extracts, rosemary extract, and estradiol reduced the number of stained cells almost completely. The activity of SA-β-gal was also assessed using flow cytometry (Figure 3c,d). Typical flow cytometry data are presented in Figure 3c. Rotenone increased SA-β-gal activity 4–5-fold compared to the control, whereas the treatment compounds reduced it to only about 30% above the control (Figure 3d). Although the extent of rotenone-induced increase in SA-β-gal activity was not the same in both methods, the results still suggest that nutritional compounds and estradiol can reduce the progression of rotenone-induced cellular senescence.

### 3.4. Rotenone Increases MMP1 and Reduces Collagen 1a1 Secretion Whereas Pre-Treatment with Carotenoids, Polyphenols, and Estradiol Reverses These Effects

The balance between MMPs and collagen protein level is important for skin elasticity and the prevention of wrinkles. Therefore, MMP-1 and pro-collagen 1a1 secretion from the dermal fibroblasts was measured in the media using specific ELISA assays. While rotenone increased MMP1 secretion by almost 2-fold, tomato and rosemary extracts and estradiol reduced this rotenone-induced rise almost to the basal level (Figure 4a). In contrast, rotenone decreased pro-collagen 1a1 secretion by almost 50%, and the tested compounds increased this secretion (Figure 4b). Collagen 1a1 mRNA expression was measured to determine if the reduction in the pro-collagen level was a result of the reduced expression or only resulted from increased degradation due to the higher MMP1 level. It was found that rotenone reduced collagen 1a1 expression by nearly 60%, and that the phytonutrients and estradiol significantly increased the collagen 1a1 mRNA (Figure 4c).

### 3.5. Mitochondrial Dysfunction following Exposure to Rotenone and Recovery by Dietary Compounds and Estradiol

Since the oxidative stress induced by rotenone and the cell damage which followed were partially corrected by the phytonutrients and estradiol, it was important to measure the mitochondrial activity to assess if this was also corrected. Mitochondrial and non-mitochondrial respiration parameters were measured using a Seahorse XF Analyzer. The cells were pre-incubated for 24 h with the treatment compounds or vehicle and then with rotenone, together with the compounds, for 3 h or 24 h, with similar results at both time points. Next, the cells were washed for 1 h according to the Seahorse protocol to remove the rotenone and treatment compounds, and the OCR and ECAR were analyzed. After measuring the basal OCR, cells were exposed sequentially to four modulators of oxidative phosphorylation (oligomycin, FCCP, and antimycin/rotenone), and the OCR was measured after the application of each modulator, enabling the sequential measurement of the ATP-linked OCR, maximal respiration, and spare capacity. Treatment with rotenone alone resulted in a substantial reduction in the OCR, which remained low during the subsequent stages of the analysis (Figure 5a). This means that all measured parameters in the rotenone-treated cells were very low, as can be seen in Figure 5b–e. Pre-treatment with the phytonutrients and estradiol protected the mitochondria from rotenone-induced mitochondrial dysfunction, showing basal OCR values that were significantly higher, reaching 50–60% of the control value (Figure 5b). The effects of the compounds on the ATP-linked OCR were examined after adding oligomycin, where the difference between the basal OCR and the OCR after oligomycin represents the ATP-linked OCR (Figure 5c). Pre-treatment with the compounds reversed the effect of rotenone to about 50% of the control. The reduction in ATP formation by rotenone was confirmed by directly measuring the ATP cellular level (Figure 5f). In the presence of rotenone alone, this level was reduced to about 40% of the control, while pre-treatment with the protective compounds restored it to about 70% of the control. The effects on glycolysis were measured by the ECAR and showed a significant increase in this rate by rotenone, as the cells were using glycolysis to compensate for the reduced mitochondrial ATP production (Figure 5g). Pre-treatment with the phytonutrients and estradiol, except for rosemary extract, restored the ECAR to the basal level. The reason for the absence of an effect of the rosemary extract is not clear and should be explored. The enhancing effect of pre-treatment with the dietary compounds and estradiol on the maximal mitochondrial respiration measured in the presence of FCCP is shown in Figure 5d, where the maximal respiration was restored to about 80% of the control, indicating improved mitochondrial function. The spare respiratory capacity that reflects the ability of the mitochondria to respond to an increased energy demand was restored to almost 100% of the control (Figure 5e).

### 3.6. Increased Activation of ARE/Nrf2 by the Combinations of the Phytonutrients or Estradiol with Rotenone

The results reported in the previous sections indicate that during pre-incubation with phytonutrients and estradiol, the cells increase their antioxidant defense capacity, which leads to decreased ROS levels and the protection of the dermal fibroblasts from adverse rotenone-induced effects. This increased antioxidant capacity may be caused by the activation of the ARE/Nrf2 transcriptional activity, which increases the transcription of antioxidant enzymes, an activity that was determined through a reporter gene assay. As expected, the transcription system was activated by rosemary extract, tomato extracts, and estradiol (Figure 6a) from 3- to 4-fold in the absence of rotenone. Rotenone alone slightly increased the activity by almost 3-fold. In the presence of rotenone, transcription system activation by the phytonutrients and estradiol was significantly higher than that of the compounds or rotenone alone, or even higher than the sum of the effects of each compound alone (Figure 6a). To clearly present the increased effect of this combination on ARE/Nrf2 transcriptional activity, the sum of the effects of the rotenone and rosemary extract is shown in Figure 6b, where the combined effect is 15-fold and is twice higher than the sum of the effects of rotenone and rosemary extract alone. To verify that rotenone increases ARE/Nrf2 transcriptional activity, and that higher activity is reached by its combination with the phytonutrients and estradiol, the protein level of NQO1, one of the antioxidant enzymes induced by this system, was detected using Western blot. Indeed, rotenone alone slightly increased the level of the NQO1 protein, and in the presence of the dietary compounds and estradiol, the protein level was significantly higher than that of the rotenone alone (Figure 6c,d).

### 3.7. Activation of ARE/Nrf2 Is Required for Protection from Rotenone-Induced Damage by Dietary Compounds and Estradiol

To investigate whether the activation of the ARE/Nrf2 transcription system is crucial for protection by the dietary compounds and estradiol, two inhibitors of the ARE/Nrf2 pathway, OTA and ML385, were used. As can be seen in Figure 7a,b, OTA or ML385 treatment at different concentrations inhibited the rotenone- and rosemary extract-induced ARE/Nrf2 transcriptional activity. The following results were from experiments performed with 25 µM OTA or 10 µM ML385. The protection, as measured by changes in cytosolic and mitochondrial ROS level, cellular ATP, cell number, and MMP-1 and pro-collagen secretion, was reduced in the presence of both inhibitors (Figure 7), whereas the effects of the treatment compounds in the absence of the inhibitors were similar to those shown in Figure 1, Figure 2, Figure 4 and Figure 5. In the presence of each inhibitor, only a small reduction in cytosolic ROS was obtained by pre-incubation with the treatment compounds (Figure 7c,d). A similar small reduction was found when mtROS was measured in the presence of OTA (Figure 7e). In addition, almost no increase in ATP level was detected when the cells were treated with the dietary compounds and estradiol in the presence of OTA (Figure 7f). These reduced effects on mtROS and ATP level in the presence of OTA suggest that the activation of the ARE/Nrf2 transcription system is necessary for an increase in mitochondrial function by these compounds. Moreover, ARE/Nrf2 activity is also necessary for the prevention of the other indications of rotenone-induced damage (Figure 7g–l). The cell number was not increased by the protective compounds in the presence of the inhibitors (Figure 7g,h). Only a partial and mostly statistically insignificant reduction in MMP-1 secretion (Figure 7i,j) and no increase in pro-collagen 1a1 secretion (Figure 7k,l) were attained by the phytonutrients and estradiol when cells were treated with the inhibitors.

## 4. Discussion

In the current study, we examined the harmful effects of endogenously generated oxidative stress in human dermal fibroblasts, which resulted from mitochondrial dysfunction induced by rotenone. Since mitochondrial function is associated with skin health, its dysfunction can lead to skin aging [43], and thus the use of rotenone may be a good model for this process (see Figure 1a). In the current study, rotenone blocked mitochondrial respiration almost completely, as is evident from a reduction in the OCR, including the ATP-linked OCR. This was associated with reduced cellular ATP, increased mitochondrial and cytosolic ROS, increased cell senescence and apoptotic cell death, increased MMP1, and reduced collagen secretion. While the reduction in mitochondrial activity by rotenone treatment was much more pronounced than that which was expected during cellular aging (see Figure 1a), the resulting robust effect on ROS and cell activity enabled significant effects of the protective compounds to be measured and the mechanisms involved in the damage and in protection to be analyzed. The association between mitochondrial ROS and the reduction in mitochondrial metabolism has previously been shown as a reduction in the mitochondrial membrane potential and oxidative phosphorylation [44]. The link between mitochondrial dysfunction and senescence was previously reported in human fibroblasts having mtDNA deletion or treated with respiration inhibitors [17,22]. In an in vivo study in mitochondrial SOD2 deficient mice, the lack of this enzyme caused increased mtROS in the epidermis, and this led to impaired mitochondrial activity and increased cellular senescence, similar to that observed in aged mouse skin [45].

Wrinkles and a loss of skin elasticity, among the disturbing phenotypes of skin aging, result from increased MMPs and reduced collagen production, which cause lower collagen levels [43] (see Figure 1a). In the current study, such changes in collagen metabolism were found following a reduction in mitochondrial activity. These outcomes are supported by a study on mitochondrial disease, myoclonic epilepsy, and ragged red fiber syndrome that occur due to genetic mutations in mtDNA which impair mitochondrial respiration, induce oxidative stress, and, in skin fibroblasts, increase MMP1 expression [46]. Additionally, depletion in mice mtDNA decreased oxidative phosphorylation in skin samples due to the instability of the electron transport complexes that reduced the expression of the MMP inhibitor TIMP1 and increased the expression of MMP2 and MMP9 [47]. In UV-exposed NHDF cells, a decreased production of collagen fibers, due to their lower secretion, was shown to be associated with the low quality of the mitochondria, which resulted in ATP reduction [48]. The lower ATP level that was correlated with increased mitochondrial ROS caused less secretion of type I collagen and fibrillin-1, which is similar to the collagen secretion results in our study.

Carotenoids, polyphenols, and estradiol protect dermal fibroblasts from the damaging effects of rotenone. These compounds lowered the ROS level, improved mitochondrial function, decreased senescence, reduced MMP1 secretion, and increased the secretion of collagen. Estradiol and the dietary compounds increased the activity of the ARE/Nrf2 transcription system cooperatively with rotenone, which suggests that this protection may be explained by the activation of this antioxidant defense system (see Figure 1b). We have previously shown that carotenoids, polyphenols, and estradiol activated ARE/Nrf2 in dermal fibroblasts, which was associated with ROS reduction and protection from H_2_O_2-_induced damage [26]. In addition to its role in activating this transcription system, it was further shown that estradiol can increase the activity of antioxidant enzymes such as hemeoxygenase-1 via an estrogen receptor α-dependent mechanism [49].

The importance of ARE/Nrf2 for cell protection was suggested by experiments using the Nrf2 inhibitors, ML385 [50] and ochratoxin A (OTA) [51]. The inhibition of ARE/Nrf2 prevented these protective effects by more than 50%, which indicates that this is one of the main mechanisms for reducing cytosolic and mitochondrial ROS, and that ARE/Nrf2 activity is also important for the improvement of mitochondrial function, and leads to prevention of the damaging effects of rotenone (see Figure 1b). In a study on ex vivo human skin, OTA blocked the anti-apoptotic effect of external DNA and bacterial biofilm, showing the importance of ARE/Nrf2 in preventing UVB-induced damage [52]. The Nrf2 inhibitor ML385 blocked the protective effect of carnosic acid, the polyphenol found in rosemary extract, against ferroptotic cell death in PC12 cells [53]. In H9C2 myocardial cells exposed to hypoxia/reoxygenation, ginseng polyphenol reversed the process of increasing ROS and apoptotic cell death, and this was also blocked by ML385 [54]. These results show that, similar to skin cells, the mechanism for reducing ROS and preventing cell death in these nerve and myocardial cell lines is the ARE/Nrf2 antioxidant system. Nrf2 was also suggested to be important for preventing senescence, as Nrf2 knockdown in human embryonic fibroblasts resulted in premature senescence [55].

In the current study, the addition of estradiol and phytonutrients improved mitochondrial respiration and increased ATP levels. The positive effect of estradiol to improve mitochondrial respiratory function, by increasing Complex I and III activities, was demonstrated in murine skeletal muscle [56]. In PC-12 cells and brain mitochondria from both female and male rats, estradiol was found to decrease mitochondrial superoxide by increasing manganese superoxide dismutase activity [57]. Estradiol was also shown to increase the mitochondria membrane potential of human dermal fibroblasts and keratinocytes under H_2_O_2_-induced oxidative stress that, like in the current study, was associated with reduced ROS and protection from cell death [34]. Decreased mitochondrial respiration occurs in Friedreich’s ataxia (FRDA) and causes a reduction in ATP production. In skin fibroblasts from a human FRDA donor, that were under oxidative stress, the protective effects of various estrogens were studied [41,42]. The estrogens increased cell viability, oxidative phosphorylation, and intracellular ATP, and prevented the collapse of the mitochondrial membrane potential. These protective effects were not related to any estrogen receptor activity and were associated with the phenolic structure and antioxidant properties of estradiol and the other estrogen-like compounds. Thus, it is important to examine the role of estrogen receptors in the protective effects of estradiol, as were found in the current study. It should be noted that the effective concentration of estradiol in these FRDA studies was 100 nM, which is ten times higher than the concentration we used in the current study.

The effects of phytonutrients in improving mitochondrial respiration are supported by previous studies that used other experimental models. For example, astaxanthin improved fatty acid utilization in murine skeletal muscle by increasing the protein amount of mitochondrial oxidative phosphorylation components, and in C2C12 cells, it increased mitochondrial biogenesis and enhanced mitochondrial oxidative respiration [58]. In addition, lycopene prevented testicular lesions induced by aflatoxin in mice by decreasing oxidative stress and increasing mitochondrial biogenesis and membrane potential [59]. In a study on endotoxin-induced damage to primary neonatal rat ventricular myocytes, the diterpene alkaloid songorine improved mitochondrial respiration by upregulating the mitochondrial genes involved in the electron transport chain and fatty acid β-oxidation, and prevented endotoxin-induced septic heart injury in mice [60]. The protective effects on mitochondrial respiration and biogenesis were dependent on the activation of ARE/Nrf2, and Nrf2 deficiency diminished this protection. In the current study, the improvement observed in mitochondrial activity, which similarly depends on ARE/Nrf2 activation, may also be associated with increased mitochondrial biogenesis; however, this issue should be examined further.

In this study, phytonutrients and estradiol reduced ROS levels and MMP1 secretion, increased collagen secretion, and decreased cell death by reducing caspase 3 activity and apoptosis. The effect of carotenoids on the apoptosis pathway in human keratinocytes has been previously shown with astaxanthin, which decreased ROS production and reversed the UVB-induced increased expression of apoptosis proteins, including BAX, caspase 3, and PARP, resulting in increased cell viability [61]. Additionally, caspase 3 expression was reduced by the topical application of lycopene, which suppressed UVB-induced apoptosis in murine skin [62]. In another study, the polyphenol resveratrol inhibited caspase 3 and 9 expression in both HaCaT keratinocytes and skin in mice [63]. Other research found that in HaCaT keratinocytes and dermal fibroblasts, estradiol helped protect against oxidative stress by decreasing the ROS level, resulting in increased cell viability and pro-collagen I synthesis [64]. Borrás et al. suggested that estradiol has a direct protective effect on the mitochondria as it decreased ROS production by isolated mitochondria, and maintained mitochondrial integrity and membrane potential, preventing the leakage of cytochrome c, which should reduce cell apoptosis [65]. Oxidative stress has been found to activate MMP expression [66,67], reduce the expression of pro-collagen, and decrease collagen synthesis [68]. Similar to the current study, estradiol was shown to decrease the expression of MMP1 and MMP9 in keratinocytes and dermal fibroblasts [34]. In aged human skin in vivo, the topical application of estradiol decreased MMP1 expression and increased that of type I pro collagen mRNA and protein [69]. Several studies have reported that carotenoids [70,71] and polyphenols [72,73,74] reduce the level or expression of MMPs and increase collagen protein expression. These changes in collagen metabolism should lead to increased skin elasticity and reduced wrinkling, and indeed human studies show that supplementation for 12 weeks with golden tomato extract [75] or 90 days with β-carotene [70] improved skin elasticity and firmness and reduced fine lines and facial wrinkles. The effects of dietary compounds on MMPs and collagen expression were associated with suppressing the activation of mitogen-activated protein kinases (MAPKs) and activator protein 1 (AP-1) signaling [73,76]. Thus, the involvement of these, and other signaling pathways, in mitigating rotenone-induced damage should be investigated.

## 5. Conclusions

The mitochondrial theory of aging suggests that the dysfunction of the mitochondria induces cell damage; thus, in the current study, we used rotenone to establish a model of mitochondrial dysfunction. Our findings indicate that carotenoids, polyphenols, and estradiol protect dermal fibroblasts from mitochondrial dysfunction-induced cell death and senescence and the deterioration of collagen homeostasis. Therefore, the consumption of these and similar dietary compounds may delay cellular senescence and skin aging. Moreover, the activation of the ARE/Nrf2 transcription system was shown to be an important mechanism for protecting skin cells from a reduction in mitochondrial activity. However, the results obtained in the presence of ARE/Nrf2 inhibitors indicate that this is not the only protective mechanism, and thus the role of other signaling pathways in cell damage and for skin protection should be analyzed.

## Data Availability

The raw data supporting the conclusions of this article will be made available by the authors on request.

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
