# Peer review of "ARE/Nrf2 Transcription System Involved in Carotenoid, Polyphenol, and Estradiol Protection from Rotenone-Induced Mitochondrial Oxidative Stress in Dermal Fibroblasts"

_antioxidants, 2024, doi:10.3390/antiox13081019_

Round 1

Reviewer 1 Report

Major comments

In this paper, Darawsha et al. studied how carotenoids, polyphenols, and estradiol protect dermal fibroblasts from damage caused by mitochondrial dysfunction and reactive oxygen species (ROS). The results showed that these compounds, by activating the antioxidant response element (ARE/Nrf2) pathway, can reduce ROS levels, prevent cell death, and maintain collagen production. I'd like to propose several major points that might need to be done to further improve the manuscript.

1. The authors analyzed cell death induced by Rotenone treatment using NHDF. However, the authors did not conduct a direct analysis of cellular senescence or skin aging. Therefore, it would be prudent to avoid making extensive claims about skin aging based on these results. If there are additional data that demonstrate a direct relationship, they should be included in the revised manuscript.

2. Similarly, skin aging and cellular senescence should be clearly distinguished, but the boundaries between these terms are often ambiguous and unclear in the current manuscript, necessitating improvement. For example, what does the term "Cell aging" in the Abstract refer to? It should be expressed as either cellular senescence or skin aging. However, as mentioned above, it is important to note that this paper does not directly analyze aging.

3. The Discussion section is somewhat redundant and unclear, as it repeatedly covers general background and interpretations of the study results. Please reduce the general parts and focus more concisely on the discussion of this study's findings.

4. This reviewer is somewhat concerned about the novelty of the manuscript. It is well known that Rotenone increases ROS levels. On the other hand, they have already reported that compounds like carotenoids increase Nrf2 activity under oxidative stress conditions such as H2O2 treatment (Antioxidants 2021, 10, 2023). What is the rationale for using Rotenone to specifically increase mitochondria-derived ROS in this study?

Minor points:

1. Figure 1: Mitochondrial ROS level Mitochondrial superoxide level

2. In Figure 5a, Nrf2 activity increases with RE and other single treatments. Does this mean that ROS levels also increase with these compounds alone? If so, could the effects of these compounds be considered hormesis effects?

3. In Figure 5d, if analyzing the expression of proteins downstream of Nrf2, it is absolutely necessary to include a Western blot for Nrf2 protein itself. It is especially important to clarify when Nrf2 protein increases after compound addition and for how many hours it remains expressed.

4. The left side of Figure 6c appears to be the same analysis as Figure 1b, but the shape of the bar graph is significantly different.

5. Scheme 1a: Mitochondrial dysfunction?

6. Scheme 1a: What do the percentages in the diagram represent?

7. In Scheme 1b, it is indicated that Carotenoids and Polyphenols directly suppress ROS, but there is no explanation or evidence for this.

8. Conclusions, L628: cell aging cell damage?

9. Conclusions, L629: What does "these processes" refer to?

10. Conclusions: Once again, please be cautious with the use of the term "aging." This study does not analyze aging, nor does it directly examine cellular senescence.

Author Response

Please see the attached PDF

Major comments

In this paper, Darawsha et al. studied how carotenoids, polyphenols, and estradiol protect dermal fibroblasts from damage caused by mitochondrial dysfunction and reactive oxygen species (ROS). The results showed that these compounds, by activating the antioxidant response element (ARE/Nrf2) pathway, can reduce ROS levels, prevent cell death, and maintain collagen production. I'd like to propose several major points that might need to be done to further improve the manuscript.

  1. The authors analyzed cell death induced by Rotenone treatment using NHDF. However, the authors did not conduct a direct analysis of cellular senescence or skin aging. Therefore, it would be prudent to avoid making extensive claims about skin aging based on these results. If there are additional data that demonstrate a direct relationship, they should be included in the revised manuscript.

We conducted experiments to demonstrate a relationship between rotenone treatment and senescence, using SA-β-gal, the most widely used biomarker for senescent cells. These experiments were not completed before the original submission of the manuscript and, thus, were not included. Following the reviewer’s suggestion to add such information, we completed the experiments and now present them in Section 3.3 and in Figure 3. The section and figure numbers, following Section 3.3, were changed accordingly. We do know that using just one marker of senescence does not constitute complete proof for this state; however, the results do suggest that rotenone increases the number of senescent cells, and the extracts/estradiol reduced them almost to the basal level.  

  1. Similarly, skin aging and cellular senescence should be clearly distinguished, but the boundaries between these terms are often ambiguous and unclear in the current manuscript, necessitating improvement. For example, what does the term "Cell aging" in the Abstract refer to? It should be expressed as either cellular senescence or skin aging. However, as mentioned above, it is important to note that this paper does not directly analyze aging.

The reviewer is correct that these two terms were not always used in the right context, and we changed them (lines 10, 26, 605, 751). Specifically, the term "cell aging" in the abstract was changed to "skin aging" as skin health is mentioned at the end of this sentence.   

  1. The Discussion section is somewhat redundant and unclear, as it repeatedly covers general background and interpretations of the study results. Please reduce the general parts and focus more concisely on the discussion of this study's findings.

We detected some parts in the Discussion (about 20 lines) that can be defined as “general background”, and we removed these.

  1. This reviewer is somewhat concerned about the novelty of the manuscript. It is well known that Rotenone increases ROS levels. On the other hand, they have already reported that compounds like carotenoids increase Nrf2 activity under oxidative stress conditions such as H2O2 treatment (Antioxidants 2021, 10, 2023). What is the rationale for using Rotenone to specifically increase mitochondria-derived ROS in this study?

The rationale for using rotenone to specifically increase mitochondria-derived ROS (mtROS) is that mitochondrial dysfunction and the resulting mtROS are associated with senescence and aging. We understand from this and other comments that this concept was not clear enough in the Introduction section, so we added additional text to make this clearer (lines 56–62, 67–71, 95–99). The senescence results that are included in the revised manuscript allowed us to consider it further in the Discussion section (lines 616–621 and 678–680). Regarding novelty, we want to emphasize that additional novelty is found in showing that ARE/Nrf2 activity is necessary for protecting the cells from the rotenone-induced damage.

Detail comments

 Minor points:

  1. Figure 1: Mitochondrial ROS level → Mitochondrial superoxide level

Writing “Mitochondrial ROS level” is justified because although superoxide is the immediate radical formed with rotenone, it can be converted to H2O2 by mitochondrial SODs (see for example, the paper by Sena and Chandel who use this term: Physiological roles of mitochondrial reactive oxygen species. Molecular Cell 2012, doi:https://doi.org/10.1016/j.molcel.2012.09.025)

  1. In Figure 5a, Nrf2 activity increases with RE and other single treatments. Does this mean that ROS levels also increase with these compounds alone? If so, could the effects of these compounds be considered hormesis effects?

The fact that carotenoids and polyphenols activate ARE/Nrf2 is well known, and such publications, including our own, are cited in the manuscript. This has nothing to do with an hormesis effect. A hormesis effect might be of the stressors (ROS), or the protector compounds. It is known that ROS has a hormesis effect, as indicated in the Introduction, but the reviewer is asking about the protecting compounds. A hormesis effect of the protectors can happen, and it is known for many antioxidants, but it is not relevant in the current study because the concentrations used were not high enough to cause damage.    

  1. In Figure 5d, if analyzing the expression of proteins downstream of Nrf2, it is absolutely necessary to include a Western blot for Nrf2 protein itself. It is especially important to clarify when Nrf2 protein increases after compound addition and for how many hours it remains expressed.

We agree that any additional information, such as Western blot for Nrf2 protein will add to the validity of the reporter gene and the NQO1 results. Such results were shown in many previous publications with these and similar compounds and, thus, we do not think that this is necessary here, especially, as the main point of the study is that activation of ARE/Nrf2 is required for the protection, as was shown with its inhibitors. This by itself is another indication that the system was activated and, thus, the results of the reporter gene and NQO1 should be enough to show this activation.  

  1. The left side of Figure 6c appears to be the same analysis as Figure 1b, but the shape of the bar graph is significantly different.

The experiments shown on the left side of Figure 6c (7c in the revision) were done about two years after the experiments shown in Figure 1b. Although the shape of the bar graph is not the same, the implication of the results is not different. In both cases, there is a large (but not equivalent) increase in cytosolic ROS by rotenone and a reduction with the extracts/estradiol. For some reason, the cells were more sensitive to the treatment compounds at that stage, but the conclusions from both figures are still valid.   

  1. Scheme 1a: Mitochondrial dysfunction?

The scheme was modified so that “mitochondrial dysfunction” is shown on the inhibitory red lines, which should make this clearer. The legend was also modified to make it clearer  

  1. Scheme 1a: What do the percentages in the diagram represent?

The scheme and its legend were modified to make it clearer. The percentages that were an inaccurate estimate were replaced by “high” and “low,” which were also added to the legend.

  1. In Scheme 1b, it is indicated that Carotenoids and Polyphenols directly suppress ROS, but there is no explanation or evidence for this.

We agree with the reviewer that the direct suppression is incorrect at the cellular level. The red arrow showing the direct effect on ROS was introduced by mistake and was removed.

  1. Conclusions, L628: cell aging → cell damage?

This has been changed.

  1. Conclusions, L629: What does "these processes" refer to?

This was changed.  "these processes" was replaced by “mitochondrial dysfunction.”

  1. Conclusions: Once again, please be cautious with the use of the term "aging." This study does not analyze aging, nor does it directly examine cellular senescence.

In line 760 in the conclusion, the term “damage” (induced by mitochondrial dysfunction) was replaced by the detailed parameters (cell death and senescence, and deterioration of collagen homeostasis) that were changed by the treatment compounds. As these parameters are associated with various manifestations of skin aging, we think that it is now justified to add to the following sentence that these compounds “may delay cellular senescence and skin aging.”

Reviewer 2 Report

This manuscript is an extension of a previous study from the same lab that looked at the effects of the same extracts/compounds (red and yellow tomato extracts, rosemary extract, estradiol) on H2O2 toxicity in normal human dermal fibroblasts. In this study, the authors test the effects of the extracts/compounds on rotenone toxicity. They use many of the same read-outs as in the previous study but also include assays for mitochondrial function since rotenone is an inhibitor of Complex I of the electron transport chain. The analyses are clearly and logically presented and the results strongly support the idea that the extracts/estradiol can prevent the effects of rotenone on the cells. In addition, the authors use two different Nrf2 inhibitors that act in distinct ways to block Nrf2 activation which very much strengthens their assertion that activation of Nrf2 is critical to the protective effects of the extracts/estradiol. However, the authors don’t clearly explain in the Introduction why they chose to use rotenone since it is more frequently used as a model of Parkinson’s disease rather than skin aging. The authors need to reframe the Introduction to make it clear that the overall goal of this study was to test the extracts/estradiol in a model of mitochondrial dysfunction which has been implicated in skin aging. Indeed, a recent paper which should be cited (PMID37394047) specifically describes using rotenone to model aging in dermal fibroblasts. There are a number of points that should also be addressed by the authors..

1.        Materials: what were the sources of rotenone and the Nrf2 inhibitors?

2.        Determination of Cell Number: it is clear from Figure 2f-h that the extracts/estradiol increase cell number. However, it seems as if the cell survival data were calculated as a percentage of untreated cells. It should be a percentage of cells treated with the extracts/estradiol alone.

3.        Fig. 1: How was the concentration of rotenone used in this and the other studies in this manuscript chosen? Also, in the figure legend, the extracts are listed with the concentration of a single compound. Does this mean that the concentration of the extract used in the assay was based on the concentration of that compound in the extract? Please clarify. Also, the authors need to explain how they chose these specific concentrations of extracts/estradiol for these studies.

4.        Fig. 2 legend: In f, g, and h, the $ symbol in the control seems to indicate a difference between cells not treated with extract/estradiol and cells treated with extract/estradiol rather than a difference between control and rotenone. Please clarify.

5.        In lines 374, 382 and 384 the authors state that the extracts/estradiol “reversed” or “restored” various aspects of mitochondrial dysfunction. However, since the authors used a pretreatment paradigm, they cannot claim reversal or restoration, only prevention. Please revise.

6.        Fig. 4 f and g: The authors need to address the observation that the rosemary extract strongly increased the ECAR in the presence of rotenone in contrast to the tomato extracts or estradiol. Why would this extract increase ECAR since it also robustly prevented the reduction of OCR? At the least, the authors should speculate based on the scientific literature on what pathways specific to this extract could be responsible for this effect.

7.        Fig. 5a legend: The description doesn’t include the treatment with rotenone. Please add.

8.        Fig. 5d does not address the question that the authors are asking. The authors need to show the effects of the extracts/estradiol alone on the levels of NQO1 since they are arguing that the combination of rotenone and the extracts/estradiol has a synergistic effect on Nrf2 activation. It is not possible to determine that without the additional data requested.

9.        Line 535: please change “proven” since nothing is truly proven in scientific research.

10.  Lines 582-583: The authors do not show that the protective effects of the extracts/estradiol on mitochondrial respiration are dependent on Nrf2 activation. Either include these experiments or drop this statement.

11.  Lines 630-631: While the authors data suggest that the extracts can delay skin aging, they do not show this. Please rephrase this sentence.

Author Response

Please see the attached PDF

Reviewer 2

 Major comments

This manuscript is an extension of a previous study from the same lab that looked at the effects of the same extracts/compounds (red and yellow tomato extracts, rosemary extract, estradiol) on H2O2 toxicity in normal human dermal fibroblasts. In this study, the authors test the effects of the extracts/compounds on rotenone toxicity. They use many of the same read-outs as in the previous study but also include assays for mitochondrial function since rotenone is an inhibitor of Complex I of the electron transport chain. The analyses are clearly and logically presented and the results strongly support the idea that the extracts/estradiol can prevent the effects of rotenone on the cells. In addition, the authors use two different Nrf2 inhibitors that act in distinct ways to block Nrf2 activation which very much strengthens their assertion that activation of Nrf2 is critical to the protective effects of the extracts/estradiol. However, the authors don’t clearly explain in the Introduction why they chose to use rotenone since it is more frequently used as a model of Parkinson’s disease rather than skin aging. The authors need to reframe the Introduction to make it clear that the overall goal of this study was to test the extracts/estradiol in a model of mitochondrial dysfunction which has been implicated in skin aging. Indeed, a recent paper which should be cited (PMID37394047) specifically describes using rotenone to model aging in dermal fibroblasts. There are a number of points that should also be addressed by the authors.

We thank the reviewer for drawing our attention to the paper by da Cruz et al. We used this reference, together with another, to reframe the Introduction (lines 56–62 and 67–71), and to explain why we used rotenone in a model of mitochondrial dysfunction which has been implicated in skin aging. We also clarify that our main goal was to test the protective effects of the extracts/estradiol in this model and to study the involved mechanisms (lines 95–99).

Detail comments

  1. Materials: what were the sources of rotenone and the Nrf2 inhibitors?

The sources are introduced in lines 132–134.

  1. Determination of Cell Number: it is clear from Figure 2f-h that the extracts/estradiol increase cell number. However, it seems as if the cell survival data were calculated as a percentage of untreated cells. It should be a percentage of cells treated with the extracts/estradiol alone.

As indicated by the reviewer, the number of cells treated with extracts/estradiol alone was higher than the control, but because the % reduction in cell number by rotenone was calculated from the control untreated cells, the positive change by the protecting compounds is clearer when the presented results are calculated from the same value of the control. As the survival results (the right-hand group of bars in Figure 2e and f in the revised manuscript) are shown alongside the results of the extracts/estradiol alone, without rotenone, it allows the reader to see the difference from rotenone alone and also to compare the survival bars to the results of each compound without rotenone.

  1. Fig. 1: How was the concentration of rotenone used in this and the other studies in this manuscript chosen? Also, in the figure legend, the extracts are listed with the concentration of a single compound. Does this mean that the concentration of the extract used in the assay was based on the concentration of that compound in the extract? Please clarify. Also, the authors need to explain how they chose these specific concentrations of extracts/estradiol for these studies.

The concentration of rotenone used in the study was determined in cell number experiments that were not described. This information is now introduced in Methods Section 2.3, lines 151–154.

When presenting results with the extracts, the concentrations given are those of the main active component in each extract. This information was added in Section 2.1. (Materials, lines 123–126).

In order to answer the last comment in point 3, the order of the panels in Fig. 2 was changed so that it starts with the dose response panels. A dose response graph of GTE, that was missing in the original submission, was also completed and added. As a result of these major changes, the all of Section 3.2 and the legend to Fig. 2 were rewritten. The reason for using the specific concentrations of the extracts/estradiol in other experiments is now explained based on the dose response results (lines 343–350).  

  1. Fig. 2 legend: In f, g, and h, the $ symbol in the control seems to indicate a difference between cells not treated with extract/estradiol and cells treated with extract/estradiol rather than a difference between control and rotenone. Please clarify.

This comment is correct for the revised Fig. 2a–d and also for Fig. 2e and f and for Figs. 5 and 6. In all these legends, the sentence was changed so that the “$” symbol shows the significant difference from the control without rotenone and the treatment compounds (lines 401, 509–510, 548–549).

  1. In lines 374, 382 and 384 the authors state that the extracts/estradiol “reversed” or “restored” various aspects of mitochondrial dysfunction. However, since the authors used a pretreatment paradigm, they cannot claim reversal or restoration, only prevention. Please revise.

The reviewer is correct that without specifying that the effects result from a pre-treatment, the phrases “reversed” or “restored” are incorrect and confusing. In the first occurrence (original line 374), the sentence was changed to: “Pre-treatment with the compounds reversed the effect” (lines 480–481).  In the second occurrence (original line 382), the sentence begins with “Pre-treatment with the phytonutrients and estradiol,” so there is no need to change it. In the third occurrence (original line 384), “pre-treatment” was added at the beginning of the sentence: The enhancing effect of pre-treatment with the dietary compounds and estradiol on the maximal…..” (line 491). As implied from comment 5, if “pre-treatment” precedes the phrases “reversed” or “restored,” there is no confusion about the meaning of the results and, thus, we think that there is no need to use the phrase “prevention.” However, we replaced “reversal” with “prevention” in other locations – in line 669 and in the legend to scheme 1b.

  1. Fig. 4 f and g: The authors need to address the observation that the rosemary extract strongly increased the ECAR in the presence of rotenone in contrast to the tomato extracts or estradiol. Why would this extract increase ECAR since it also robustly prevented the reduction of OCR? At the least, the authors should speculate based on the scientific literature on what pathways specific to this extract could be responsible for this effect.

We did not find any reasonable explanation for the result showing that rosemary extract did not reduce the rotenone-induced increase in ECAR. Actually, a recent paper (https://doi.org/10.3390/ijms25094983) showed that carnosic acid, which is the main active component in rosemary extract, suppressed glycolysis, which is opposite to this result. We added a sentence to convey this notion (lines 489-490).

  1. Fig. 5a legend: The description doesn’t include the treatment with rotenone. Please add.

Rotenone was added (line 541)

  1. Fig. 5d does not address the question that the authors are asking. The authors need to show the effects of the extracts/estradiol alone on the levels of NQO1 since they are arguing that the combination of rotenone and the extracts/estradiol has a synergistic effect on Nrf2 activation. It is not possible to determine that without the additional data requested.

The sentence describing NQO1 results in Figure 6 c and d, arrives immediately following the sentence showing the higher cooperative activation of ARE/Nrf2 transcriptional activity. This proximity gives the impression that cooperative activity is measured also on NQO1 protein level. However, this was not how the results were described (lines 532-535). It was only stated that “in the presence of the dietary compounds and estradiol, the protein level was significantly higher than that of the rotenone alone”. Analysis of protein level by Western blot is a semi-quantitative method with low sensitivity and thus, such cooperative activity is difficult to show with this method. To prevent confusion, the introductory sentence for the NQO1 results was changed (lines 529-531) to clarify what was verified by this analysis.

  1. Line 535: please change “proven” since nothing is truly proven in scientific research.

“proven” was changed to “suggested by experiments”, line 664.

  1. Lines 582-583: The authors do not show that the protective effects of the extracts/estradiol on mitochondrial respiration are dependent on Nrf2 activation. Either include these experiments or drop this statement.

We include experiments to show that the protective effects of the extracts/estradiol on the mitochondria are dependent on Nrf2 activation (lines 559, 564-570, 585-587 and Figure 7 e and f). The effect of the inhibitors on mitochondrial respiration per se, using the Seahorse technology, was not tested because it requires more than a month to perform but the revision had to be submitted within about two weeks. However, the results of the effects on mtROS and ATP level in the presence of OTA suggest that activation of the ARE/Nrf2 transcription system is necessary for the improvement of mitochondrial function.

  1. Lines 630-631: While the authors data suggest that the extracts can delay skin aging, they do not show this. Please rephrase this sentence.

According to the suggestion of reviewer 1, we added experiments showing that rotenone increases cell senescence and pretreatment with the extracts/estradiol reduce it (Section 3.3 and Figure 3). In addition, in the sentence preceding lines 630-631 (in the original submission) we detailed the parameters (cell death and senescence and deterioration of collagen homeostasis (lines 763-764)) that were changed by the treatment compounds. As these parameters are associated with various manifestations of skin aging, we think that the phrase “may delay cellular senescence and skin aging” in the current sentence (line 765) is justified.
